# Antibiofilm and Antiquorum Sensing Potential of *Lactiplantibacillus plantarum* Z057 against *Vibrio parahaemolyticus*

**DOI:** 10.3390/foods11152230

**Published:** 2022-07-27

**Authors:** Xiangpeng Han, Qingying Chen, Xingguo Zhang, Xiaolan Chen, Dongsheng Luo, Qingping Zhong

**Affiliations:** 1Guangdong Provincial Key Laboratory of Food Quality and Safety, College of Food Science, South China Agricultural University, Guangzhou 510642, China; hanxp1996@163.com (X.H.); qingyingchen@stu.scau.edu.cn (Q.C.); zhangxingguoscau@163.com (X.Z.); chenxiaolan@stu.scau.edu.cn (X.C.); 2College of Tobacco Science, Henan Agricultural University, Zhengzhou 450002, China

**Keywords:** *Vibrio parahaemolyticus*, biofilm, quorum sensing, virulence factors, *Lactiplantibacillus plantarum* Z057

## Abstract

*Vibrio parahaemolyticus* is a widespread foodborne pathogen that causes serious seafood-borne gastrointestinal infections. Biofilm and quorum sensing (QS) are critical in regulating these infections. In this study, first, the ability of *Lactiplantibacillus plantarum* Z057 to compete, exclude, and displace *V. parahaemolyticus* biofilm was evaluated. Then, the inhibitory effects of *L. plantarum* Z057 extract (Z057-E) on *V. parahaemolyticus* biofilm and QS were explored from the aspects of biofilm biomass, metabolic activity, physicochemical properties, extracellular polymer matrix content, QS signal AI-2 activity, biofilm microstructure, and the expression levels of biofilm and QS-related genes. Results showed that *L. plantarum* Z057 effectively inhibited biofilm formation of *V. parahaemolyticus* and interfered with the adhesion of *V. parahaemolyticus* on the carrier surface. In addition, the Z057-E could significantly reduce the biofilm biomass, metabolic activity, hydrophobicity, auto-aggregation ability, swimming and swarming migration diameter, AI-2 activity, extracellular polysaccharide (EPS), and extracellular protein content of *V. parahaemolyticus*. Fluorescence microscope and scanning electron microscope (SEM) images demonstrated that the Z057-E could efficiently inactivate the living cells, destroy the dense and complete biofilm architectures, and reduce the essential component of the extracellular polymer matrix. Real-time fluorescence quantitative PCR revealed that the Z057-E treatment down-regulated the expression of flagellum synthesis-related genes (*flaA*, *flgM*), EPS, and extracellular protein synthesis-related genes (*cpsA*, *cpsQ*, *cpsR*, *ompW*), QS-related genes (*luxS*, *aphA*, *opaR*), and hemolysin secretion-related genes (*toxS*, *toxR*) of *V. parahaemolyticus*. Thus, our results suggested that *L. plantarum* Z057 could represent an alternative biocontrol strategy against foodborne pathogens with anti-adhesive, antibiofilm, and antiquorum sensing activities.

## 1. Introduction

*Vibrio parahaemolyticus* is a common Gram-negative motile bacterium living in various aqueous environments, especially in estuary water, freshwater, and marine water [1,2]. It is frequently associated with foodborne disease outbreaks, usually following the consumption of insufficiently cooked or raw seafood [3]. The factors responsible for pathogenesis in *V. parahaemolyticus* are multiple virulence factors such as biofilm formation, flagellum, adhesion, lipopolysaccharides, secretion systems, and cytotoxins, which are regulated by a typical cell-to-cell communication system called quorum sensing (QS) [4]. QS is a cell signaling mechanism that responds to fluctuations in cell population density through the secretion of autoinducers (AIs) [5]. The signal molecule AIs accumulate with cell density. When AIs reach a critical concentration, they can bind to transcription factors and regulate functional gene expression [6]. AI-2 is a crucial signal molecule for interspecific communication of *V. parahaemolyticus*, and its synthesis and secretion are regulated by the cascade of *luxS*, *opaR*, and *aphA* genes [7]. Therefore, strategies developed to inhibit the QS provide a novel alternative for preventing and controlling *V. parahaemolyticus* biofilm.

Biofilm is a sessile microbial community in which bacterial cells are encased in a self-produced matrix composed of extracellular polysaccharides (EPS), proteins, lipids, and eDNA [8]. It is the leading cause of seafood spoilage or transmission of foodborne pathogens. Moreover, *V. parahaemolyticus* possesses polar flagellum for swimming [9], and the presence of lateral flagellum helps in swarming [10], both of which are essential functions of biofilm formation and adhesion during the establishment of infection [11]. Bacterial biofilm formation is a momentous survival mechanism. The extracellular polymer matrix of biofilm can provide a protective barrier for pathogenic bacteria when biofilm-associated cells are exposed to unfavorable environments, including antibacterial agents, nutritional deficiency, and high salinity [12,13]. In food processing environments, *V. parahaemolyticus* survives by forming biofilms on shrimp, crab, oysters, mussels, and the surface of stainless steel, glass, rubber, polytetrafluoroethylene, and other materials [8,14]. Consequently, *V. parahaemolyticus* biofilm has attracted considerable attention because of its pathogenicity, antibiotic resistance, and ability to contaminate seafood worldwide [15].

Currently, antibiotics and disinfectants are mainly used to kill foodborne pathogenic bacteria in clinics and food processing. However, conventional disinfectants cannot effectively eliminate pathogenic biofilms and leave them on the surface of food processing equipment, resulting in product contamination [16]. In addition, large amounts of antibiotics used for controlling pathogen infection have resulted in an increase in multiple drug-resistant bacteria due to the ineffectiveness of antibiotics therapy [17]. Therefore, it is essential to explore novel antibiofilm agents or therapeutic approaches to interfere with biofilm development and prevent antibiotic resistance of pathogenic bacteria [18]. Most lactic acid bacteria (LAB) species are generally recognized as safe (GRAS), which are isolated frequently from various food sources, including dairy, vegetables, fruits, meat, and traditional fermented foods [19]. LAB as potential probiotics can be considered as substitutes for disinfectants and antibiotics in clinics and food processing [20]. Compared with conventional disinfectants and antibiotics, naturally derived LAB are considered safer, more people-friendly, and more effective [21]. Moreover, as food supplements or components of the microbiota, LAB can directly inhibit pathogens in a strain-specific manner by competing for binding sites or nutrients, binding pathogens, and impeding pathogenic internalization [22]. More importantly, LAB metabolites such as bacteriocins, organic acids, biosurfactants, and EPS directly inhibit the pathogens’ biofilm formation and their QS activity in long-term periods [23].

In our previous study, the LAB strains with potential antibacterial and antibiofilm effects were screened, and *Lactiplantibacillus plantarum* Z057 exhibited excellent antibacterial ability against *V. parahaemolyticus*. In addition, the strain also showed a high level of cell auto-aggregation, co-aggregation with *V. parahaemolyticus*, cell surface hydrophobicity, and biofilm formation ability (the data will be published in another paper). On this basis, the potential ability of *L. plantarum* Z057 to compete, exclude, and displace pathogens during biofilm formation of *V. parahaemolyticus* was investigated. Moreover, the antibiofilm and anti-QS abilities and mechanisms of *L. plantarum* Z057 extract (Z057-E) were evaluated by monitoring the changes in biomass, metabolic activity, extracellular polymer matrix, and structure of *V. parahaemolyticus* biofilm. Additionally, the effects of Z057-E on QS signal activity, hydrophobicity, auto-aggregation ability, and motility of *V. parahaemolyticus* were investigated, and the transcription levels of biofilm formation-associated genes and QS-related genes were also examined. This study aimed to clarify the potential value of *L. plantarum* Z057 and analyze its antibiofilm and anti-QS activities, which provided a sound basis for further developing Z057-E as a natural preservative or antibiofilm agent in the food industry and other fields.

## 2. Materials and Methods

### 2.1. Bacterial Strain and Culture Conditions

*L**. plantarum* Z057, preserved in our laboratory, was cultured in Mann Rogosa Sharpe (MRS, Guangdong Huankai Microorganism Technology Co., Ltd., Guangzhou, China) broth at 37 °C for 12 h. *Vibrio parahaemolyticus* ATCC 17802 and *Vibrio harveyi* BB170 were obtained from the Guangdong Culture Collection Centre of Microbiology (Guangzhou, China). *V. parahaemolyticus* ATCC 17,802 was inoculated into tryptic soy broth (TSB, Guangdong Huankai Microorganism Technology Co., Ltd., Guangzhou, China) containing 3% (*w/v*) NaCl and incubated at 37 °C for 12 h at 150 rpm. *V. harveyi* BB170 was grown aerobically in Autoinducer Bioassay medium (AB, Guangdong Huankai Microorganism Technology Co., Ltd., Guangzhou, China) at 30 °C for 12 h at 90 rpm. Based on a method previously described by Xie et al. [8] and Shangguan et al. [17], all strains maintained at −80 °C were activated for three generations before being used in the experiment.

### 2.2. Co-Culture of L. plantarum Z057 and V. parahaemolyticus

The competition, exclusion, and displacement assays were performed according to the methodology reported by Woo and Ahn [24] and Perez-Ibarreche et al. [25]. For the competition assay, an equal volume of *L. plantarum* Z057 (10^8^ CFU mL^−1^) and *V. parahaemolyticus* (10^6^ CFU mL^−1^) was added to a 24-well plate co-cultured in TSB for 48 h at 37 °C. For the exclusion assay, *L. plantarum* Z057 biofilm was initially settled on the bottom of a 24-well plate by incubation in MRS at 37 °C for 48 h. Then, the plate was rinsed with sterile saline to remove non-adherent cells and immediately inoculated with *V. parahaemolyticus* (10^6^ CFU mL^−1^) and incubated in TSB for 48 h at 37 °C. For the displacement assay, *V. parahaemolyticus* biofilm was initially formed on the bottom of a 24-well plate by incubation in 3%NaCl-TSB at 37 °C for 48 h. Then, non-adherent cells were removed by a rinse with sterile saline. *L. plantarum* Z057 (10^8^ CFU mL^−1^) was added to the prepared *V. parahaemolyticus* biofilm and incubated for 48 h at 37 °C. The cultures of *V. parahaemolyticus* were used as controls. For enumeration, *V. parahaemolyticus* planktonic and biofilm cells were collected from the 24-well plate by rinsing with sterile saline. Proper dilutions were plated on TCBS agar (Guangdong Huankai Microorganism Technology Co., Ltd., Guangzhou, China) and incubated at 37 °C for 48 h.

### 2.3. Determination of the Minimum Inhibitory Concentration (MIC)

*L. plantarum* Z057 culture was centrifuged (4 °C, 8000× *g* rpm, 15 min) followed by filtering (0.22 µm filter, Millipore, Billerica, MA, USA) to obtain cell-free supernatant (CFS). The CFS of *L. plantarum* Z057 was extracted with ethyl acetate, evaporated, and freeze-dried, and the dried powder, named Z057-Extrant (Z057-E), was stored at −80 °C. The MIC was determined using a published method with minor modifications [15]. Two-fold dilution of Z057-E in 3% NaCl-TSB resulted in a range of 0 to 6.4 mg/mL, which was transferred to sterile 96-well plates. Then, aliquots (4 μL) of *V. parahaemolyticus* suspension (10^8^ CFU mL^−1^) were inoculated into a medium, and the plates were incubated at 37 °C for 24 h. Each well’s optical density (OD) was measured at 595 nm using a microplate reader (Thermo Fisher Scientific, Waltham, MA, USA).

### 2.4. Biofilm Inhibition Assay

Biofilm inhibition assay was performed in 96-well plates as previously described, with some modifications [8]. Briefly, 4 μL of *V. parahaemolyticus* suspensions (10^8^ CFU mL^−1^) were inoculated into 3%NaCl-TSB containing Z057-E at tested concentrations (0, 0.4, 0.8, and 1.6 mg/mL). After incubation at 37 °C for 24 h, the crystal violet staining assay was used to quantify the biofilm biomass [26]. The wells were gently washed with phosphate buffer saline (PBS, pH 7.2), dehydrated for 30 min, stained with 0.1% crystal violet (Shanghai Macklin Biochemical Technology Co., Ltd., Shanghai, China) for 5 min, and then washed three times with PBS again. After drying for 30 min, the dyed biofilms were treated for 10 min with 33% (*v*/*v*) glacial acetic acid. The biofilm biomass was quantified by measuring the absorbance at 595 nm using a microplate reader (Thermo Fisher Scientific, Waltham, MA, USA).

### 2.5. Measurement of the Metabolic Activity of Biofilms

XTT reduction assay was used to analyze the metabolic activity of biofilm [27]. XTT (2, 3-bis (2-methoxy-4-nitro-5-sulfophenyl)-2H-tetrazolium-5-carboxanilide) (Sigma-Aldrich, Gillingham, UK) solution (1 mg/mL) was prepared with sterile water, mixed with menadione (Shanghai Sangon Biological Engineering Technology & Service Co., Ltd., Shanghai, China) in a ratio of 12.5:1. After washing the plate with PBS for three times, 100 μL PBS and 13.5 μL mixture were added to each well, and the plate was incubated for 2 h at 37 °C in the dark. The OD_450nm_ was measured using a microplate reader (Thermo Fisher Scientific, Waltham, MA, USA).

### 2.6. Evaluation of the Physicochemical Properties of V. parahaemolyticus Treated with Z057-E

Cell surface hydrophobicity assay was performed based on a modification of the procedure presented by Salaheen et al. [28]. Briefly, bacterial culture was grown in the absence (control) and presence of Z057-E (final concentrations of 0.4, 0.8, and 1.6 mg/mL). After incubation at 37 °C for 24 h, the cells were collected by centrifuging at 8000 rpm for 15 min, washed twice using PBS, and resuspended in PBS to maintain an OD_595nm_ (H_0_) between 0.4 and 0.6. Then, 1.5 mL of the bacterial suspension was mixed with 1.5 mL of Xylene. The mixture was vortexed perfectly for 2 min and incubated for 20 min to allow for the separation of the two phases. The absorbance of the aqueous phase was measured at 595 nm (H_t_) by a microplate reader (Thermo Fisher Scientific, Waltham, MA, USA). The hydrophobicity was calculated as a percentage using the following formula: Hydrophobicity (%) = (1 − H_t_/H_0_) × 100.

Auto-aggregation assay was conducted according to the method described previously [28], with some modifications. Bacterial culture without (control) or with Z057-E (final concentrations of 0.4, 0.8, and 1.6 mg/mL) was incubated at 37 °C for 24 h. The cells were harvested by centrifuging at 8000× *g* rpm for 15 min, washed twice with PBS, and resuspended in PBS, and OD_595nm_ was adjusted to between 0.4 and 0.6 (A_0_). The cell suspensions were then cultured for 2 h at 37 °C. After collecting the supernatants, a microplate reader (Thermo Fisher Scientific, Waltham, MA, USA) was used to determine their absorbencies at 595 nm (At). The auto-aggregation rate was calculated using the following formula: Auto-aggregation rate (%) = (1 − A_t_/A_0_) × 100.

### 2.7. Swarming and Swimming Motilities

Swarming and swimming motilities were performed as previously described by Packiavathy et al. [29]. Swarm agar medium containing glucose (1%), peptone (0.6%), yeast extract (0.2%), agar (0.5%), and Z057-E (final concentrations of 0, 0.4, 0.8, and 1.6 mg/mL) were prepared. Then, 2 μL of *V. parahaemolyticus* culture (10^6^ CFU mL^−1^) were spotted on the center of the agar plate, then incubated upright at 37 °C for 24 h. Meanwhile, swimming agar medium consisting of tryptone (1%), NaCl (0.5%), and agar (0.3%) supplemented with Z057-E (final concentrations of 0, 0.4, 0.8, and 1.6 mg/mL), and 2 μL of *V. parahaemolyticus* culture (10^6^ CFU mL^−1^) were spot inoculated onto the agar plate’s center. The plates were then incubated for 24 h at 37 °C in an upright position, and swarming and swimming migration zones were recorded.

### 2.8. Autoinducer-2 (AI-2) Bioassay

A bioluminescence assay was carried out to determine whether AI-2 production in *V. parahaemolyticus* was affected by Z057-E. AI-2 was detected using the *V. harveyi* BB170 biosensor described previously [30], with a few modifications. Briefly, 5 mL of 3%NaCl-TSB containing various concentrations of Z057-E (0, 0.4, 0.8, and 1.6 mg/mL) and 2% mixed *V. parahaemolyticus* culture (10^8^ CFU mL^−1^) were added to the test tubes. After 24 h of incubation at 37 °C, 150 rpm, the culture was centrifuged at 12,000× *g* rpm for 10 min. The supernatants were then filtered using a 0.22 μm filter and stored at −80 °C. The reporter strain *V. harveyi* BB170 was cultivated aerobically in AB medium at 30 °C for 12 h at 100 rpm. The culture was diluted 1:5000 (*v*/*v*) in AB medium, and 180 μL was added to each well of a 96-well black opaque plate. Then, 20 μL of supernatant from each tested sample was added to the wells. The mixtures were incubated at 30 °C for 3 h at 100 rpm in a shaker. In this study, the positive controls (180 μL of culture with 20 μL of the non-treated supernatant sample) were used, and the negative control merely contained a sterile medium. Light production from *V. harveyi* BB170 was measured using a microplate analyzer (Molecular Device, San Jose, CA, USA).

### 2.9. Analysis of Extracellular Polymer Matrix in Biofilm

The overnight culture of *V. parahaemolyticus* (10^8^ CFU mL^−1^) was inoculated in 2 mL fresh 3%NaCl-TSB containing sterile coverslips (12 mm× 12 mm) in 24-well plates, treated with Z057-E at 0, 0.4, 0.8, and 1.6 mg/mL. After culturing for 24 h at 37 °C, the extracellular polymer matrix was extracted by the ultrasonic method according to the protocol of Zhang et al. [31]. The contents of EPS and protein were determined by the phenol-sulfuric acid method [8] and the Coomassie brilliant blue method [31].

### 2.10. Visualization of the Biofilms by Fluorescence Microscopy

The effects of Z057-E on *V. parahaemolyticus* biofilms were verified by the method described by Xie et al. [8]. Biofilms were cultured on sterile coverslips (12 mm× 12 mm) in 24-well plates with Z057-E at different concentrations (0, 0.4, 0.8, and 1.6 mg/mL). After incubating at 37 °C for 24 h, the biofilm specimens were gently washed twice with PBS to remove planktonic cells. Using a LIVE/DEAD BacLight bacterial viability kit (Proteintech, Rosemont, IL, USA), 5 μL of mixed staining solution (2 μM Calcein-AM and 1.5 μM PI) was added and treated in the dark at room temperature for 15 min. Meanwhile, 5 μL of fluorescein isothiocyanate (FITC)-Con-A (Sigma-Aldrich, Gillingham, UK) was added to the PBS-washed biofilm specimens and maintained at 4 °C for 30 min. Finally, the coverslips were gently fixed on the clean glass slides and observed under a fluorescence microscope (ZEISS, Oberkochen, Germany).

### 2.11. Visualization of the Biofilms Using Scanning Electron Microscopy (SEM)

SEM observation was performed as described by Yan et al. [30], with some modifications. The biofilms on the coverslips were fixed with 2.5% glutaraldehyde for 24 h at 4 °C and serially dehydrated by a graded series of ethanol (30, 50, 70, 80, 90, and 100%) for 10 min each time. The coverslips were sputter-coated with gold under vacuum and visualized using SEM (FEI, Hillsboro, OR, USA).

### 2.12. Analysis of the Gene Expression Using RT-qPCR

The overnight culture of *V. parahaemolyticus* was treated without (control) or with 0.8 mg/mL Z057-E at 37 °C for 24 h, and the cells were collected by washing three times with sterile PBS. Then, the total RNA was extracted using the total bacterial RNA extraction kit (Shanghai Sangon Biological Engineering Technology & Service Co., Ltd., Shanghai, China) according to the manufacturer’s protocol. After measuring the RNA concentration using a nucleic acid and protein spectrophotometer (Merinton Instrument, Ann Arbor, MI, USA), cDNA was synthesized using the RevertAid™ First Strand cDNA synthesis kit (Thermo Fisher Scientific, Waltham, MA, USA) following the manufacturer’s instructions. The cDNA samples were preserved at −20 °C. The gene-specific primers and 16S rRNA housekeeping gene primers (synthesized by Sangon Biotech Co., Ltd., Shanghai, China) are listed in Table 1. RT-qPCR was performed in a 25 μL reaction volume using SYBR Green Master Mix (Takara) on a StepOne™ Real-Time PCR system (ABI, Foster, CA, USA). The cycling parameters included an initial denaturation at 95 °C for 3 min, followed by 45 cycles of 95 °C for 5 s, 60 °C for 30 s, and primer extension at 72 °C for 30 s. The samples were analyzed in triplicate, and the relative gene expressions were analyzed with the 2 ^-ΔΔCT^ method, as previously described [28,32].

### 2.13. Statistical Analysis

All experiments were performed independently in triplicate, and all results were presented as the mean ± standard deviation (SD). One-way analysis of variance was used to compare the value differences (*p* < 0.05) using SPSS software (Version 25.0, IBM, Armonk, NY, USA). All graphical evaluations were performed using GraphPad Software (Version 8.0, GraphPad, San Diego, CA, USA).

## 3. Results

### 3.1. L. plantarum Z057-Mediated Inhibition on V. parahaemolyticus Biofilm

The inhibitory effects of the strain on pathogenic biofilm were evaluated by competition, displacement, and exclusion, which could simulate the conditions in bacterial communities. The ability of *L. plantarum* Z057 planktonic cells to competitively inhibit the biofilm formation by *V. parahaemolyticus* is shown in Figure 1a. After co-culture for 48 h, the number of planktonic and biofilm cells of *V. parahaemolyticus* reduced by 0.89 log CFU mL^−1^ and 1.54 log CFU mL^−1^, respectively, compared with the single-strain culture of *V. parahaemolyticus* (control). The ability of *L. plantarum* Z057 biofilm cells to exclude biofilm formation by *V. parahaemolyticus* is presented in Figure 1b. The *L. plantarum* Z057 biofilm that had previously developed on the abiotic surfaces impeded the formation of *V. parahaemolyticus* biofilms. Specifically, the number of planktonic and biofilm cells of *V. parahaemolyticus* decreased by 1.16 log CFU mL^−1^ and 1.61 log CFU mL^−1^ after 48 h, respectively. The ability of *L. plantarum* Z057 cells to displace *V. parahaemolyticus* biofilm cells is shown in Figure 1c. The treatment of *L. plantarum* Z057 planktonic cells was able to displace the pre-established *V. parahaemolyticus* biofilm and reduced the count of planktonic cells by 1.36 log CFU mL^−1^ and biofilm cells by 0.95 log CFU mL^−1^ after 48 h. Compared to competition and exclusion assays, the biofilm formation by *V. parahaemolyticus* was less inhibited in displacement assay.

### 3.2. Effects of Z057-E on Biofilm Formation of V. parahaemolyticus

The MIC of Z057-E against *V. parahaemolyticus* was determined as 3.2 mg/mL, and the three concentrations of Z057-E (0.4, 0.8, and 1.6 mg/mL) that had no significant influence on *V. parahaemolyticus* growth were selected as sub-MICs for the following experiment. The inhibition effects of Z057-E at different concentrations on biomass and metabolic activity of *V. parahaemolyticus* biofilm are presented in Figure 2. Results showed that Z057-E significantly inhibited biofilm biomass and metabolic activity, and inhibitory rates of biofilm biomass and metabolic activity increased gradually with the increase in Z057-E concentration. When *V. parahaemolyticus* was treated with Z057-E (final concentrations of 0.4, 0.8, and 1.6 mg/mL) for 12 h, biofilm biomass decreased by 29.44%, 52.08%, and 63.59%, and the metabolic activity decreased by 62.15%, 65.28%, and 83.07%, respectively. After 24 h, Z057-E still presented a strong inhibitory effect on *V. parahaemolyticus* biofilm, reducing biofilm biomass by 20.07%, 42.29%, and 54.56% and metabolic activity by 31.17%, 46.19%, and 77.53%, respectively. Nevertheless, it is worth noting that the inhibitory effects of Z057-E on biomass and metabolic activity of *V. parahaemolyticus* biofilm gradually weakened with the duration of action.

### 3.3. Effects of Z057-E on the Auto-Aggregation and Cell Surface Hydrophobicity of V. parahaemolyticus

Auto-aggregation and cell surface hydrophobicity positively correlated with bacterial adhesion and biofilm formation [35]. When Z057-E was added, auto-aggregation and hydrophobicity of *V. parahaemolyticus* decreased significantly. In the auto-aggregation assay, the untreated bacterial cells exhibited high auto-aggregation, whereas, in the presence of Z057-E, the values were reduced to 13.70% at 0.8 mg/mL Z057-E and 4.81% at 1.6 mg/mL (Figure 3a). In the hydrophobicity assay, treatment with 0.8 mg/mL of Z057-E reduced 19.81% hydrophobicity compared with the control. When Z057-E reached 1.6 mg/mL, the hydrophobicity rate decreased by 35.86% (Figure 3b). Therefore, it was speculated that Z057-E inhibited biofilm formation by reducing the aggregation and cell surface hydrophobicity of *V. parahaemolyticus*.

### 3.4. Effects of Z057-E on the Swarming and Swimming Abilities of V. parahaemolyticus

Swarming and swimming migrations play a critical role in the QS-mediated factor expression and biofilm formation of *V. parahaemolyticus* [36]. As shown in Figure 4, the control group showed the largest migration diameters for swarming and swimming, which were 11.51 ± 0.44 mm and 55.58 ± 3.08 mm, respectively. In the experimental group, Z057-E effectively reduced the swarming and swimming motilities of *V. parahaemolyticus* in a dose-dependent manner. At the tested concentrations (0.4, 0.8, and 1.6 mg/mL), Z057-E significantly decreased the diameters of the swarming areas by 38.70%, 40.90%, and 49.04%, respectively, while the same treatment significantly reduced the swarming diameters by 24.58%, 49.52%, and 54.70%, respectively. The above results indicated that the Z057-E interfered with the adhesion and motility of *V. parahaemolyticus* flagellum.

### 3.5. Effect of Z057-E on the Activity of Signal Molecule AI-2 of V. parahaemolyticus

Autoinducer-2 (AI-2) is considered a universal language for intraspecies and interspecies communication, and it is related to the formation of *V. parahaemolyticus* biofilm [7]. As shown in Figure 5, Z057-E could significantly and dose-dependently inhibit the synthesis of AI-2. Moreover, following the exposure to 0.8 and 1.6 mg/mL of Z057-E, the AI-2 productions were reduced by 56.52% and 66.69%, respectively.

### 3.6. Effect of Z057-E on EPS, Extracellular Protein of V. parahaemolyticus Biofilm

EPS and extracellular protein are the main components of bacterial biofilm, and perform many functions, including promoting adhesion to surfaces, constructing and maintaining biofilm structures, and protecting cells during antibacterial treatment [12]. Therefore, inhibiting or reducing their secretion is crucial for controlling biofilm formation. As shown in Figure 5, with the increase in Z057-E concentration, the extracellular protein and EPS and extracellular protein excretions in the biofilm were significantly decreased. After 0.8 mg/mL of Z057-E treatment, EPS and extracellular protein contents were reduced by 62.70% and 50.25%, respectively. After being treated with 1.6 mg/mL of Z057-E, EPS and extracellular protein contents were decreased by 78.76% and 51.91%, respectively. Consequently, it was speculated that Z057-E directly acted on the bacterial cells after destroying the biofilm structure.

### 3.7. Fluorescence Microscope Visualization of Biofilm

A fluorescence microscope was applied to visualize the live and dead cells in *V. parahaemolyticus* biofilm. The living cells of *V. parahaemolyticus* were labeled with Calcein-AM (green fluorescence), whereas dead cells were stained with propidium iodide (red fluorescence). At the same time, the EPS of biofilm was stained by FITC Con-A to produce green fluorescence. In the absence of Z057-E, the living cells appeared to be relatively intensive, and only a few scattered dead cells could be observed in *V. parahaemolyticus* biofilm (Figure 6a1,b1). With 0.8 mg/mL of Z057-E treatment, the activity of biofilm cells changed significantly, which could be observed by the decrease of living cells and the increase of dead cells (Figure 6a2,b2). Moreover, a large number of dead cells appeared in the field of vision when the biofilm was treated with 1.6 mg/mL of Z057-E (Figure 6a3,b3). Simultaneously, we observed that the fluorescence intensity of EPS in the untreated biofilm was higher, especially in the middle position, indicating the thicker area of the biofilm (Figure 6c1). After the treatment, the images of the two experiment groups presented similar trends: the green fluorescence intensity was remarkably decreased, revealing a notable reduction in EPS content (Figure 6c2,c3).

### 3.8. Scanning Electron Microscope (SEM) Visualization of the Changes of Biofilm Structures

Visualization of biofilm structure and organization through SEM revealed the antibiofilm effects of Z057-E on *V. parahaemolyticus*. In Figure 7a, the control samples showed regularly organized network structures, and a compact extracellular polymer matrix surrounded the bacterial cells. However, the architectural integrity of the biofilm and the cell aggregation of cells were significantly changed following exposure to 0.8 mg/mL of Z057-E (Figure 7b). With increasing concentrations of Z057-E, the biofilm structure was disrupted entirely, with only a few dispersed single cells (Figure 7c).

### 3.9. Z057-E Modulated Expressions of Biofilm-Related Genes and QS-Related Genes in V. parahaemolyticus

The effects of Z057-E treatment on the expressions of biofilm-related genes and QS-related genes were quantified by real-time PCR. As indicated in Figure 8, Z057-E treatment down-regulated the expression levels of flagellum regulation-related genes (*flaA* and *flgM*) by 18.83% and 47.15% compared to the control (Figure 8a). The expression levels of QS-related genes (*luxS*, *aphA*, *opaR*) were decreased by 36.68%, 29.67%, and 25.13%, respectively (Figure 8b). Moreover, after Z057-E treatment, the *cpsA*, *cpsR*, *cpsQ*, and *ompW* genes, which regulated the syntheses of EPS and extracellular proteins, were also significantly down-regulated by 34.02%, 35.27%, 59.00%, and 53.84%, respectively (Figure 8c). Meanwhile, the Z057-E treatment down-regulated the expression levels of toxin production-related genes including *toxR* and *toxS* by 15.37% and 58.89%, respectively (Figure 8a).

## 4. Discussion

In this study, competition, exclusion, and displacement assays were carried out to evaluate the inhibitory effects of *L. plantarum* Z057 on the biofilm formation of *V. parahaemolyticus* in complex bacterial communities. In the competition assay, during the co-culture with *L. plantarum* Z057, the number of *V. parahaemolyticus* sessile cells was reduced. Similarly, co-cultivation with LAB strains could influence *Listeria monocytogenes* adhesion and biofilm formation [37]. The substances produced by LAB, such as organic acids, bacteriocins, and biosurfactants, could inhibit the growth and adhesion of pathogenic bacteria on the biological or abiotic surface or even provoke cell separation from the biofilm structures [25,38]. In addition, LAB could inhibit pathogenic biofilm formation via other strategies, such as competition for space, nutrients, and adhesion sites [39]. In exclusion assay, biofilm formation by *V. parahaemolyticus* was prevented by a preformed *L. plantarum* Z057 biofilm on carrier surfaces, and *V. parahaemolyticus* sessile cells decreased significantly compared with the control group. The inhibitory effect of LAB on pathogenic biofilm is mainly due to its production of inhibitory growth factors. The EPS produced by LAB could prevent the adhesion of pathogenic bacteria on host tissues or inorganic surfaces, promote the dispersion of biofilm cells into plankton, and alter the balance between biofilm and planktonic populations [40]. In displacement assay, the inhibition effect was less than in competition and exclusion assays. The result was consistent with the previous report that the probiotic-mediated effect of post-treatment was less than that of pretreatment [24,25]. The adhesion and colonization of pathogenic bacteria on the carrier surface was the initial stage of biofilm formation. Therefore, it was an effective strategy to prevent and control pathogenic bacterial pollution by interfering with the adhesion of pathogenic bacteria on both biotic and abiotic surfaces or reducing their adhesion ability.

The extracellular polymer matrix of biofilm is primarily composed of EPS, proteins, and eDNA, which are the first line of defense against preservatives, sterilizing agents, antibiotics, etc. [41]. The destruction of the extracellular polymer matrix allows antibacterial drugs to effectively target the live cells in biofilm and inactivate them [42]. In the present study, Z057-E significantly reduced the metabolic activity, the EPS, and the extracellular protein contents of *V. parahaemolyticus* biofilm. The fluorescence microscopy and SEM results were consistent with the above data, demonstrating that Z057-E effectively decreased the number of viable cells, increased the dead cell counts, destroyed the dense and complete structure of the biofilm, and transformed the biofilm cells from an aggregated state to a dispersed one. In the study of Onbas et al. [43], the biomass, metabolic activity, EPS, and drug resistance of biofilm were reduced by co-incubation with the cell-free extract of *L. plantarum* F-10. Similarly, Wang et al. [44] defined that metabolites of *L. plantarum* inhibited the *Bacillus licheniformis* biofilm development by reducing live/dead cell counts, metabolic activity, and EPS content of biofilm adhered on stainless steel and glass surfaces. Moreover, the observed results of the antibiofilm activities of LAB metabolites were also determined by Kanmani et al. [45] and Rana et al. [46].

The outer membrane proteins (OMPs) are key players in nutrition absorption and interactions with the environment and host tissues [47]. The transcription of the *ompW* gene in *V. parahaemolyticus* is associated with biofilm development [32]. The VPA1403-1412 (*cpsA-J*) operon of *V. parahaemolyticus* is responsible for producing EPS [48]. Specifically, *cpsQ* is a cyclic diguanylate monophosphate (c-di-GMP)-binding regulatory protein, which could positively control the intracellular messenger and the *cpsA* gene transcription. The c-di-GMP is crucial in mediating bacterial biofilm formation and multidrug efflux pump expression [49]. The *cpsR* gene is also required to induce EPS-related gene expression [50]. In our study, RT-qPCR results showed that Z057-E significantly down-regulated the expression levels of EPS-related genes (*cpsA*, *cpsQ*, *cpsR*) and extracellular protein-related genes (*ompW*). These findings further demonstrated that Z057-E could reduce the content of the extracellular polymer matrix, penetrate the collapsed biofilm, disrupt the membrane permeability of biofilm cells, and lead to cell apoptosis. This finding is in agreement with the results of Li et al. [33], who showed that acidic electrolyzed water effectively inhibited the expressions of EPS-related genes (*cpsA*, *cpsQ*, *cpsR*) of *V. parahaemolyticus*, prevented the production of the extracellular polymer matrix, and destroyed the three-dimensional structures of biofilms. Similarly, Sun et al. [32] reported that citral inhibited *V. parahaemolyticus* biofilm formation by disrupting biofilm-associated protein biosynthesis and suppressing OMP-associated gene expression (*ompW*).

The biofilm formation of *V. parahaemolyticus* is closely related to the movement ability of the flagellum. In aid of flagellum, bacterial cells perform initial adhesion and colonization to assist biofilm formation [51]. *flaA* and *flgM* genes are vital for regulating the biosynthesis and structure of *V. parahaemolyticus* flagellum. Specifically, FlaA, specialized polar flagellin, regulates the production of flagellar filaments, resulting in bacteria swimming and infecting host cells [52]. FlgM, an anti-factor, could mirror the potential advantages of swarming motility, increasing the number of flagellum filaments and switching the swimmer cells to swarmer cells [11]. Sun et al. [32] demonstrated that sub-MICs of citral (12.5 μg/mL) could down-regulate the flagellum synthesis-related genes (*flaA* and *flgM*) of *V. parahaemolyticus*, reducing the swimming and swarming abilities by 47% and 50%, respectively, and interfere with the adhesion and biofilm formation of *V. parahaemolyticus*. Similarly, the crude extract from *Lactobacillus crustorum* ZHG 2-1 effectively inhibited the swarming ability of *Pseudomonas aeruginosa* [36]. In this study, after Z057-E treatment, the diameters of swimming and swarming movement and the expression levels of *flaA* and *flgM* genes decreased significantly compared with the control group, indicating that Z057-E could effectively inhibit the biosynthesis of the polar and lateral flagellum, interfering with the motility and biofilm formation of *V. parahaemolyticus*.

Adherence to host tissues is a prerequisite for pathogenic bacteria to establish infection [53]. *V. parahaemolyticus* acts on host tissue and causes pathogen infections via flagellum movement [54]. In the present study, we also found that Z057-E down-regulated the transcription level of hemolysin-related genes (*toxS* and *toxR*) of *V. parahaemolyticus*. This finding was consistent with the result of a previous study that the thymoquinone could significantly repress the adhesion, invasion, and hemolysin secretion abilities of *V. parahaemolyticus* by inhibiting the swimming and swarming motility, the expressions of flagellum and hemolysin-associated genes [53]. In addition, the auto-aggregation ability reflects bacterial cell interaction and functions as an adhesion process for integrating and establishing bacteria in biofilm communities [35]. In some cases, another vital surface physicochemical property is the cell surface hydrophobicity of microorganisms, which reflects their attachment to host tissue [28]. Our study determined the effects of Z057-E on the physicochemical properties of *V. parahaemolyticus*. The changes in auto-aggregation ability and hydrophobicity may lead to the reductions of adhesion ability and biofilm formation of *V. parahaemolyticus* during Z057-E treatment, but whether they were the determining factors needs further investigation.

The maturity of a biofilm is dictated not only by structural elements such as flagellum, pili, and EPS biosynthesis but also by regulatory mechanisms such as c-di-GMP and QS [55]. *V. parahaemolyticus* possesses two central QS systems (LuxM/N and LuxS/AI-2), which regulate biofilm formation and promote biofilm-related infection. AI-2, a dihydroxy pentanedione-derived chemical generated by LuxS-like synthases, is involved in inter-species communication in a broad range of bacteria [32]. The LuxS/AI-2 QS system is cascade regulated by two core regulatory factors (AphA and OpaR) and regulates multiple virulence-related genes, including biofilm formation, motility, bacterial colonization, EPS synthesis, and toxin secretion [34]. AphA is the major QS regulator of *V. parahaemolyticus* operating under low cell density, while OpaR is the master QS regulator under high cell density [56]. Ashrafudoulla et al. [57] found that eugenol quenched the activity of QS signal AI-2 and down-regulated expression of the *luxS* gene, thus inhibiting the adhesion and biofilm formation of *V. parahaemolyticus*. Other studies showed that the citral interfered with the QS system, effectively inhibiting motility, virulence, and biofilm formation of *V. parahaemolyticus* [32]. Similarly, Chen et al. [34] also reported that curcumin-mediated photodynamic inactivation could impede *V. parahaemolyticus* colonization, infection, and biofilm formation by suppressing the expression of QS-related genes (*aphA*, *opaR*, and *luxS*). In this study, the Z057-E efficiently inactivated the living cells and AI-2 activity, inhibited the motility and biofilm formation, degraded the main chemical composition of the extracellular polymer matrix, and negatively altered changed biofilm structures. Moreover, the Z057-E treatment down-regulated the expression of the QS-related genes (*aphA*, *opaR*, and *luxS*) of *V. parahaemolyticus*, which would impede bacterial colonization and biofilm formation. Therefore, the Z057-E was a novel LAB agent for preventing and controlling *V. parahaemolyticus* biofilm pollution. It also exhibited significant development and application potential as an anti-QS agent from LAB.

## 5. Conclusions

This study reported that *L. plantarum* Z057 exhibited excellent antibiofilm and anti-QS potential against *V. parahaemolyticus*. The antibiofilm and anti-QS mechanisms of *L. plantarum* Z057 include the following. Firstly, *L. plantarum* Z057 could significantly reduce the number of *V. parahaemolyticus* cells in plankton and biofilm by competition, displacement, and exclusion, and interfere with the adhesion of *V. parahaemolyticus* on the carrier surface. Secondly, *L. plantarum* Z057 extract (Z057-E) could destroy the biofilm barrier, directly acting on the bacterial cells and causing the reduction in metabolic activity. Thirdly, the Z057-E could inhibit the movement ability, hydrophobicity, and auto-aggregation of *V. parahaemolyticus*. Hence, we speculated that it affected the initial colonization and adhesion, thereby affecting the biofilm formation. Fourthly, the Z057-E could inhibit the QS signal molecule AI-2, EPS, and extracellular protein secretions. Lastly, the Z057-E could down-regulate the expressions of flagellum-related genes (*flaA* and *flgM*), EPS and extracellular protein-related genes (*cpsA*, *cpsQ*, *cpsR*, and *ompW*), QS-related genes (*luxS*, *aphA*, and *opaR*), and hemolysin-related genes (*ToxS* and *ToxR*) of *V. parahaemolyticus*. In conclusion, *L. plantarum* Z057 could be applied as a safe, green, and effective microbial agent to control the pollution and infection of *V. parahaemolyticus* biofilm.

## Figures and Tables

**Figure 1 foods-11-02230-f001:**
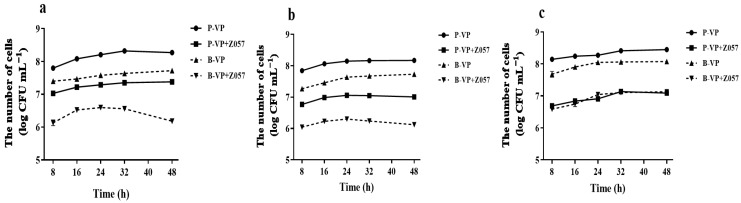
Competition (**a**), exclusion (**b**), and displacement (**c**) ability of *L. plantarum* Z057 against *V. parahaemolyticus* biofilm. P—Planktonic cells; B—Biofilm cells.

**Figure 2 foods-11-02230-f002:**
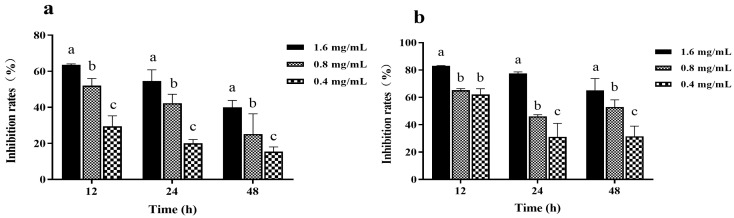
The effects of Z057-E on the biomass (**a**) and metabolic activity (**b**) of *V. parahaemolyticus* biofilm. Significant differences (*p* < 0.05) are indicated by different letters (a, b, and c).

**Figure 3 foods-11-02230-f003:**
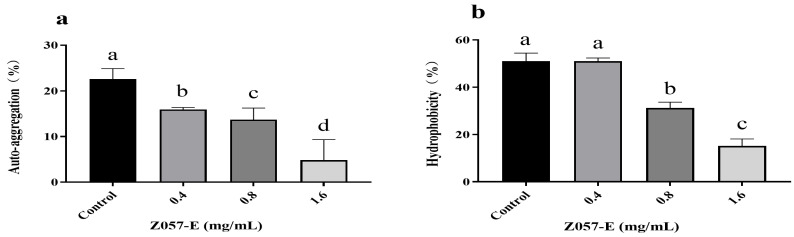
The effects of Z057-E on the auto-aggregation (**a**) and cell surface hydrophobicity (**b**) of *V. parahaemolyticus*. Significant differences (*p* < 0.05) are indicated by different letters (a, b, c, and d).

**Figure 4 foods-11-02230-f004:**
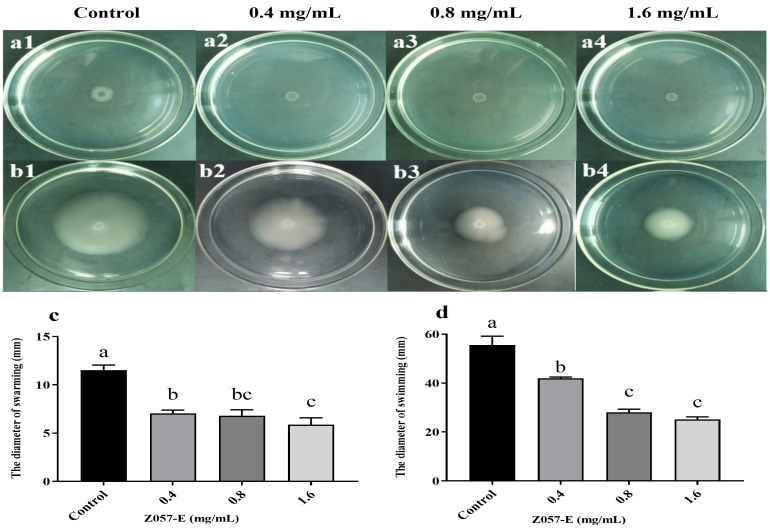
The effects of Z057-E on swarming (**a1**–**a4**,**c**) and swimming (**b1**–**b4**,**d**) abilities of *V. parahaemolyticus*. Significant differences (*p* < 0.05) are indicated by different letters (a, b, and c).

**Figure 5 foods-11-02230-f005:**
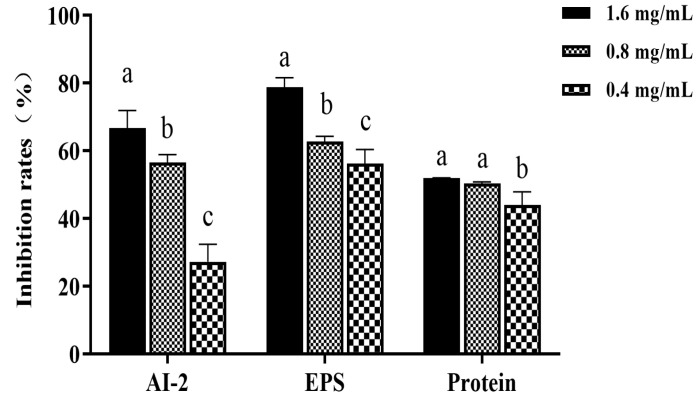
The effect of Z057-E on AI-2 activity, EPS, and extracellular protein synthesis of *V. parahaemolyticus*. Significant differences (*p* < 0.05) are indicated by different letters (a, b, and c).

**Figure 6 foods-11-02230-f006:**
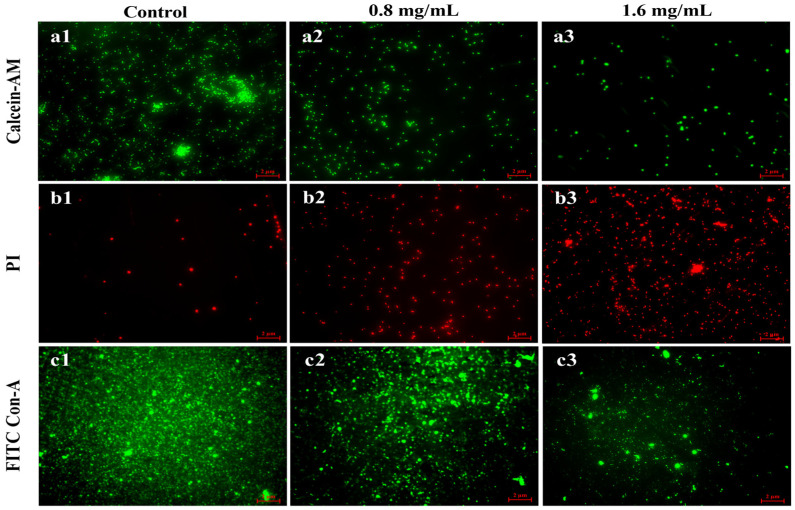
The effects of Z057-E on cells and EPS in *V. parahaemolyticus* biofilms as observed by fluorescence microscope. (**a1**,**b1**) exhibited the cell distribution of untreated biofilms, and green or red fluorescence, indicating that the cells were alive or dead. (**a2**,**b2**) represented the distribution of live and dead cells in biofilms treated with 0.8 mg/mL of Z057-E. (**a3**,**b3**) showed the distribution of live and dead cells in biofilms treated with 1.6 mg/mL of Z057-E. (**c1**–**c3**) displayed the distribution of EPS (green fluorescence) in biofilms treated with 0, 0.8, and 1.6 mg/mL of Z057-E, respectively.

**Figure 7 foods-11-02230-f007:**
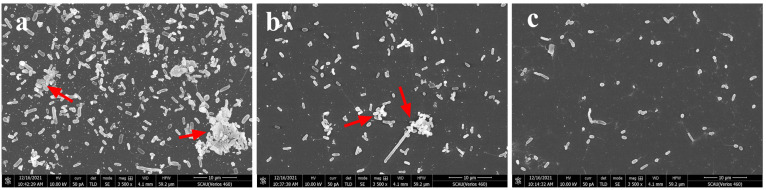
The effects of Z057-E on the structures of *V. parahaemolyticus* biofilms as observed by scanning electron microscope. (**a**–**c**) represented the structures of biofilms treated with 0, 0.8, and 1.6 mg/mL of Z057-E, respectively. The red arrows pointed to the aggregated cells.

**Figure 8 foods-11-02230-f008:**
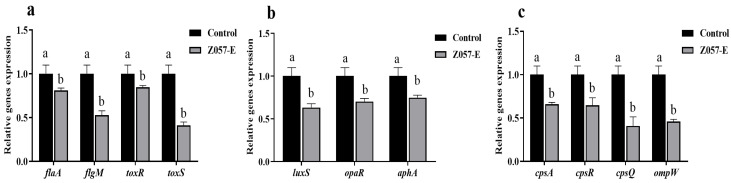
The effects of Z057-E on the expression levels of *V. parahaemolyticus* biofilm-related genes and QS-related genes. (**a**) Flagellum synthesis and hemolysin secretion-related genes. (**b**) QS-related genes. (**c**) EPS and extracellular protein synthesis-related genes. Significant differences (*p* < 0.05) are indicated by different letters (a and b).

**Table 1 foods-11-02230-t001:** Primer sequences of the RT-qPCR assay.

Genes	Primer Sequences (5′–3′)	References
*16S rRNA*	F-GCCTTCGGGAACTCTGAGACAG R-GCTCGTTGCGGGACTTAACCCAA	This study
*flaA*	F-CGGACTAAACCGTATCGCTGAAA R-GGCTGCCCATAGAAAGCATTACA	[32]
*flgM*	F-ATGCGAATTCCATGGCAGGTATAGATAATATA R-ATGCCTCGAGGCTTTTGCCTTGCAATTCGTT	[32]
*luxS*	F-GGATTTTGTTCTGGCTTTCCACTT R-GGGATGTCGCACTGGTTTTTAC	[32]
*opaR*	F-TGTCTACCAACCGCACTAACC R-GCTCTTTCAACTCGGCTTCAC	[32]
*aphA*	F-ACACCCAACCGTTCGTGATG R-GTTGAAGGCGTTGCGTAGTAAG	[33]
*cpsA*	F-GAGAGCGGCAACCTATATCG R-GCGGTCAAACAAAGGGTAAAC	[33]
*cpsR*	F-TTGGAGTCGCACTCTGGTCAA R-TGCACGCGACACACCAAGTT	[33]
*cpsQ*	F-GCCTGAAATCCTAATGCTC R-AGTGTCAGAAGGTGTATCAAC	[33]
*ompW*	F-TCGTGTCACCAAGTGTTTTCG R-CGTGGCTGAATGGTGTTGC	[32]
*toxR*	F-CAACGAAAGCCGTATACTCCTG R-CTCAAAACCTTGCTCACGCC	[34]
*toxS*	F-ATTTTTCTGAAGCGCAACTACG R-CCGTAGAACCGTGATTTAGGCT	[34]

## Data Availability

All data are reported in the article.

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
