# Peer review of "Antibiofilm and Antiquorum Sensing Potential of Lactiplantibacillus plantarum Z057 against Vibrio parahaemolyticus"

_foods, 2022, doi:10.3390/foods11152230_

Round 1

Reviewer 1 Report

This manuscript describes anti-biofilm and anti-quorum sensing potential of L. plantarum Z057 against V. parahaemolyticus. The results seemed valuable. However, there are some points to be clarified.

1.      Please explain the reason why L. plantarum Z057 was selected.

2.      Please explain whether other L. plantrum strains or lactic acid bacteria do not exhibit anti-biofilm and anti-quorum sensing potential. L. plantrum having not potential should be used as control.  

3.      It is well known that the substance produced by lactic acid bacteria, such as organic acid, bacteriocins, and biosurfactants could inhibit growth and adhesion of pathogenic bacteria and inhibit pathogenic biofilm formation. What are the new findings in this paper?

4.      How can you prove that the L. plantarum used in Fig. 6 and Fig. 7 have the same density?

5.    Please add footnotes in Fig. 6 and Fig. 7.

6.   Lactobacillus plantarum Lactiplantibacillus plantarum. Please change it throughout the entire manuscript.

Reviewer 2 Report

Dear Editor and author 

The manuscript (Anti-biofilm and anti-quorum sensing potential of Lactobacil-2 lus plantarum Z057 against Vibrio parahaemolyticus) needs  several corrects and modifications.

1-Since 2020, the names of the bacteria that belong to the lactic acid bacteria have received name changes. Why is the old name used for these bacteria in the manuscript? Correct to Lactiplantibacillus plantarum .

2-The introduction of the manuscript needs to add new reference about Vibrio bacteria in food , I suggest one reference ( Al-Sahlany, S. T. G. (2016). Effect of Mentha piperita essential oil against Vibrio spp. isolated from local cheeses. Pak. J. Food Sci, 26(2), 65-71.‏)

3-Page 2 line 96-97, The activation method of Lactiplantibacillus plantarum is an error, This bacteria needs 18–24 h for growth and activity. I suggest you to read (Niamah, A. (2019). Ultrasound treatment (low frequency) effects on probiotic bacteria growth in fermented milk. Future of Food: Journal on Food, Agriculture and Society, 7(2), Nr-103.‏).

4-The activation method of The activation method V. parahaemolyticus and V. harveyi need to add new references.

5-Page 3 line 105 and 106, Delete line 105 and 106, in here the method chapter.

6-Page 3, line 109, The method is not clear, what is the culture medium used here?

7- Many working methods do not contain references, example Determination of the minimum inhibitory concentration (MIC),   Visualization of the biofilms by fluorescence microscopy, and Visualization of the biofilms using scanning electron microscopy (SEM).

8-Page 4 line 149 and 160, Why are the phrases repeated?

9-Page 4 line 170-180, This method is unclear. Generally, in the motility of bacteria, test tubes are used, not plates.

10-Figure 6 , the picture is unclear, Please write a detailed explanation below the figure .

11-In Figure 8, I mentioned a control sample, there is no indication of it in the method of work mentioned on the page 5 line 219-234.

12-Page 13 line 496, The name of bacteria should be written in italics in all manuscript. 

Round 2

Reviewer 2 Report

Dear Editors, 

The authors made all the necessary changes to improve the manuscript, and now I recommend it for publication in its current form.